# Implementing the Technologies of Additional Impermeable Layers in a Building of the Monuments Office (Káčerov Majer) from a Sustainability Point of View

**Patrik Šťastný** [1,*]🆔, **Peter Makýš** [1], **Ivan Vavrík** [1] **and Daniel Kalús** [2]🆔

[1] Department of Building Technology, Faculty of Civil Engineering, Slovak University of Technology in Bratislava, 810 05 Bratislava, Slovakia

[2] Department of Building Services, Faculty of Civil Engineering, Slovak University of Technology in Bratislava, 810 05 Bratislava, Slovakia

* Correspondence: patrik.stastny@stuba.sk

**Abstract:** This article presents the results of an experiment that involved applying additional impermeable layers on an historical building in the area of Káčerov Majer. The building showed a high level of waterlogging, which was detected by an initial inspection and subsequent moisture measurements that involved taking a set of measurements and recording the data in a spreadsheet linked to the plan view of the building. Then, a suitable method was devised to prevent rising damp and to assist in drying out the building. Several measurements after the application of remediation interventions were used for this purpose. The interventions, together with a comprehensive restoration, were intended to contribute to the reuse of the building, and thus its sustainability from technical, environmental, functional, and economic points of view. The aim of this research was to demonstrate the degree of effectiveness of undercutting technology in combination with local application of stainless-steel sheet piling and grouting technology, as one example of sustainability, which is a very relevant topic at the moment. These technologies were intended to provide remediation of the building against rising damp which was destructive to the structure. The environment in this building was affected by moisture and was also unhealthy, which made it difficult to use the building. For these reasons, it was necessary to take radical steps, by applying invasive anti-moisture technologies. The appropriateness and effectiveness of the technologies used have been evidenced by the results published in this article that have also been supplemented by photographs before and after the application of the remediation measures. This research demonstrates the effectiveness of the interventions and the need to also implement them on certain selected buildings where there is a high level of dampness causing degradation. From a sustainability point of view, such a step is essential to preserve the life of such structures and their continued use. The results of this research clearly demonstrate the appropriateness of comprehensive restoration of selected historical buildings, complemented by invasive damp-proofing interventions, as one of the important aspects of the sustainability of deteriorating monuments, whose numbers are constantly increasing.

**Keywords:** reinstatement; undercutting; stamping of stainless-steel sheets; grouting; additional insulation; Káčerov Majer area sustainability; historical building; historical monument

## 1. Introduction

Historical buildings form an integral part of every nation's cultural heritage, which is also the case in our country. These monuments represent material heritage and also the history of our country. Many historical buildings reflect the stages of cultural development in our regions. A number of documents exist regarding the conservation and restoration of such buildings, which help to preserve this heritage. There are various methodologies that different countries have developed which should be followed for the conservation and restoration of historical buildings. From a global point of view, the Venice Charter is

considered to be an appropriate model. This international charter lays down the rules for heritage conservation. The Venice Charter briefly defines the basic concepts, conservation, and restoration procedures, etc.

Buildings gradually degrade with age or due to insufficient and neglected maintenance. Consequently, the result is their partial or even complete destruction. Therefore, it is necessary to repeatedly and comprehensively restore damaged structures using different construction technologies. Such interventions often ensure that the lifetimes of buildings are extended, which falls under the term "sustainability". More specifically, sustainability of similar buildings can be described as follows: Many historical buildings decay for many years and the materials gradually degrade, leading to their destruction. The elements of destruction form waste debris around the buildings, which creates a burden on the environment. By applying remediation measures, this problem can be completely eliminated, which is a significant benefit in terms of environmental sustainability. In addition, in terms of "visual sustainability", the renovation of buildings is very important. Deteriorated buildings often have a negative impact on their surroundings. Restoration has the effect of eliminating such a problem altogether. Economic sustainability is another point worth mentioning. If the problem is continually resolved, it leads to repeated repairs, which are financially demanding in the long term. In the case of a more radical solution, the root cause of the problem is solved. The problem (in our case, rising damp) no longer exists, and therefore, the consequences do not have to be addressed again. This leads to financial savings in the long term. The final and most important aspect of sustainability is the fact that buildings which have been rehabilitated can be reused and given a new purpose, which may not be the same as the original use. The concept of sustainability is one of the most important aspects of our future. Sustainability is a current topic in the construction sector [1,2] and also in other sectors [3,4] that are not focused on construction. This fact is confirmed by the number of articles that have been published in recent years. For example, a very interesting and beneficial article by Leila [5] described the concepts of green economy, green investment, and heritage environments, which are values and investment, quality of life, and indicators that must be considered when dealing with heritage environments and investment as an entrance to the quality of life.

This paper deals with the application of invasive remediation interventions for eliminating rising damp in structures. The interventions represent part of the complex renovation of the building mentioned below, but from the point of view of prolonging its functional period, they are one of the most important factors that will ensure its long-term sustainability from various perspectives. This, in consequence, is also linked to the re-use of the building. Moisture, in general, is one of the most significant and widespread problems in our climatic zone. The presence of water in structures affects the architectural heritage and a significant portion of historical buildings [6–8]. For example, about half of the restorations of monuments in Belgium are related to high humidity and salinization of the structures [9]. The issue of dampness has been reported as early as 1892. This problem was pointed out by Kenwood [10]. Unfortunately, this problem has not been addressed for many years, a fact supported by a study by [11] that collected data on the application of remediation technologies. It was clear from the results that the solution of remediation interventions has been very narrowly treated until recently.

Today, we know that solving the problems caused by rising damp is essential, not only for the remediation of the internal building environment, but also as a key factor in the protection of buildings. For historical buildings, addressing these problems is complemented by the heritage protection factor, which is also key from a sustainability perspective; such interventions protect the historical value of buildings, and the buildings can continue to be used for a variety of purposes.

Adding to the difficulties in rehabilitating historical structures, many of the tables used to determine the rate of dampness vary according to different standards. In addition, the classification of salinity, which is closely related to dampness, varies among different

guidelines and standards. For example, the different classifications of salinity according to WTA E-2-9-04/D, ČSN P 73 06 10, Önorm B 3355-1, etc. can only be mentioned verbally.

## 2. Current Status and Overview of the Implementation of Remediation Technologies

There are many methods of remediation described in various publications [12,13] that may not cope adequately with high levels of dampness and in some cases are not effective at all. These technologies have long been variously divided from a global point of view. R. W. Sharpe [14] divided these methods into traditional and non-traditional methods. Among the traditional methods he included technologies providing the insertion of an impermeable layer. The second group, i.e., non-traditional methods, included grouting and electroosmotic systems. A similar classification has been described in other publications. An interesting and more specific division was given in the EMERISDA project [15], where these technologies were divided into two main groups, which are presented in Figure 1.

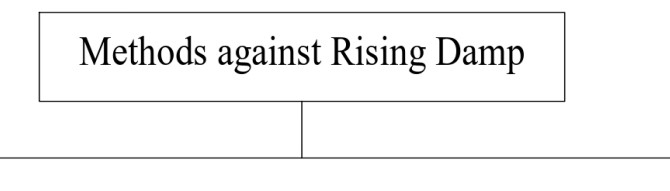

Methods against Rising Damp

**Methods which stop/limit rising damp**

1. Damp-proof course
   1.1 Mechanical interruption
   1.2 Chemical interruption
        1.2.1 injection
        1.2.2 impregnation

2. Evaporating increase
   2.1 Knapen siphons
   2.2 Drying stones
   2.3 Wall base ventilation
   2.4 Dehumidification plasters

3. Elektrokinetic
   3.1 Elektro-osmosis
        3.1.1 Active
        3.1.2 Pasive
   3.2 Others systems

**Methods which tackle symptoms (i.e. additional measures)**

1. Special plasters

2. Veneer walls

3. Drainage

**Figure 1.** Division of methods intended for the rehabilitation of structures, according to [15].

In [16], the author Makýš divided remediation technologies according to the construction-physical and implementation aspects into seven main groups (Figure 2).

1. Technologies providing ventilation:
   - creation of ventilation ducts
   - provision of ventilation by a contact channel
   - provision of ventilation through a contact slot
   - creation of a pre-wall
   - creation of hollow floors

2. Technology for creating additional impermeable layers:
   - newly lined insulation joint
   - undercutting of the masonry
   - embedding of stainless steel sheets
   - laying an additional waterproofing layer

3. Technology for creating crystalline curtains:
   - creation of hydrophobicity curtains
   - creation of sealing and hydrophobicity curtains

4. Technologies using electro-physical principles:
   - installation of galvanoosmosis devices
   - installation of passive electro-osmosis devices
   - installation of active electro-osmosis devices
   - installation of wireless dehumidification devices

5. Technology of heating the structures:
   - installation of concealed heating
   - installation of microwave drying equipment
   - implementation of hot air drying

6. Additional technologies:
   - implementation of waterproofing coatings
   - implementation of waterproofing plasters and sealants
   - implementation of remediation plasters
   - desalination of masonry

7. Related technologies:
   - creation of drainage
   - lowering the groundwater level
   - creation of vapour-permeable modifications of the surrounding area

**Figure 2.** Division of methods intended for the rehabilitation of structures, according to Makýš, O.

However, most of these technologies have not been the subject of research, and the information about them is only complementary to a general overview.

In previous years, many studies have focused on grouting technology and investigating its effectiveness. Regrettably, in only a few cases, the form of application of the grout to the structure has been reported, which makes it very difficult to compare all the results to date with those in the paper. In 1977, Sharpe, R.W., in [14], described research on five different types of masonry that had been vacuum saturated with water prior to testing. Five different materials were examined in flow tests, i.e., a soft French limestone and four different bricks. The four bricks were Stock, Fletton, Hammill, and a type of brick taken from a free-standing wall at Audley End House. Subsequently, seven different substances that applied directly to the material under study were investigated in the laboratory. Three commercially used water repellents were used in the formulation required for damp-proof course work, i.e., a silicone in white spirit, aluminum stearate in white spirit, and sodium methyl siliconate in water. All were supplied as concentrates which were diluted to the recommended concentration of 5% solids content (by weight). In addition, white spirit alone, butyl, and amyl alcohols were used as alternative higher viscosity fluids for the water immiscible injections, as well as a 5% solution of siliconate made more viscous by the addition of glycerol. The author drew a number of conclusions regarding individual substances, but stated that the results obtained under laboratory conditions did not simulate actual in situ conditions. Thus, such results could not evaluate the effectiveness of individual substances directly in the field. The author also pointed out that the effectiveness of the research would be directly confirmed or refuted by experimental work on the walls of damp structures. Studies carried out since the 1980s have reported experimental research on crystalline screens in Australia [17], or a few years later, grouting methods were directly described on the Baroque Ludwigsburg Palace, where only a slight decrease in dampness was observed after grouting application [13]. Later, van Hees et al. [18] investigated six selected samples that were first tested in the laboratory and later tested directly in practice. Specifically, these substances were applied directly by the manufacturers of each solution to the walls of Hemiksem Abbey in Belgium. After the application of the crystalline membranes, the masonry was left to dry for 2.5 years, and then measurements and sampling

were performed. The results showed only a low functionality of these screens and only a minimal tendency for the structure to dry out. In contrast, in [19], the author described the implementation of grouting at Schönbrunn Castle (Austria), where the initial values were measured in 1988 and subsequently the author measured the moisture content in 2000. The recorded values at the beginning showed a moisture content between 13.5% and 19.4%. Significant waterlogging of the building could be observed. Subsequently, when measurements were conducted in 2000, the dampness was approximately 2% in most areas. It follows that the technology demonstrated effectiveness against damp ingress in the rehabilitated structure. Unfortunately, the author lacked continuous moisture measurements after the application of the technology to represent the gradual drying of the structure after the application of grouting and to establish adequate conclusions. This would have confirmed that, above the level of grouting, the structure would have ceased to become damp. Continuous measurements would further help to rule out the influence of other factors such as environmental humidity and climatic influences.

The fact remains that a long time has passed since the introduction of grouting technology, which according to [20] dates back to the 1960s. During this time, the construction industry has evolved, as well as the technologies and materials used in it, which has resulted in an increase in the quality and efficiency of these materials and technologies that can be considered to be effective today. It should be noted that many factors come into consideration that cannot be influenced and contribute to the efficiency of the technologies. These include, for example, the porosity of the material, the type of boreholes, the method of filling, and many other factors.

The technology of indenting stainless-steel plates has not avoided some difficulties. From a technological point of view, this technology should be functional, as evidenced by the results of our research; however, the problem tends to be inexpert implementation or errors in the implementation of such technology. In particular, the joints and overlaps of the embedded sheets, through which dampness can penetrate, are problematic. This example is dealt with, among others, in [19]. The latter investigated the implementation of sheet piling on a chapel in Hollabrunn (Austria). The authors stated that moisture penetrated through weak spots at the level of the seal, i.e., the contact between the two sheets. From practice and research, the authors confirmed the demonstration of the functionality of such technologies on several buildings [21]. Unfortunately, most of the results are not yet published and will be part of future articles and studies.

The most widespread and widely used technology is undercutting technology. It is used worldwide in various types of construction. For example, a study by [22] described the use of diamond wire undercutting technology (the technology is called HIO technology) [23], which dealt with the rehabilitation of a former commercial academy building in Zrenjanin, Serbia, dating back to 1892. The detailed description of the rehabilitation of the building was divided into three steps, which were presented in detail in the paper. The authors stated that the technology eliminated 100% of the rising damp because all horizontal surfaces of the rehabilitated walls were cut as low as possible from the ground or foundation. Another application was described in [24]. In this case, the technology was used at the University of Prishtina, and the application was considered to be effective. On the contrary, in [25], the authors stated that the method in question was ineffective, mainly due to its unprofessional implementation.

## 3. Applied Technologies

Several remediation technologies were investigated by applying them, to some extent, to the building under study. The primary technology was masonry undercutting. Due to the difficulty and complexity of the building, almost all undercutting methods were used. Handsaw cutting (Figure 3a) and chainsaw cutting (Figure 3b) were used in some areas of the building, but diamond rope cutting (Figure 3c) was used in the majority of the building. Undercutting technology was implemented on approximately 95% of the building.

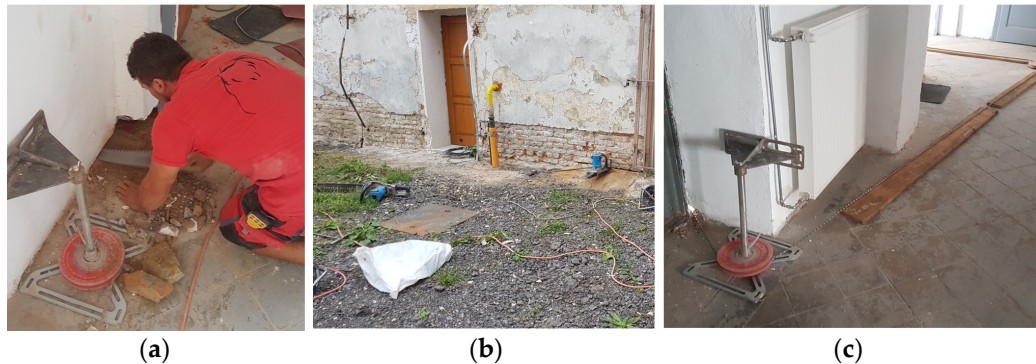

**Figure 3.** Undercutting technology: (**a**) By hand saw; (**b**) by chainsaw; (**c**) by diamond rope.

In areas where it was not possible to undercut the masonry, stainless-steel sheet piling technology was applied (Figure 4). This was implemented in minimal quantities in areas where the masonry was crumbling or where plastic elements could not be used due to the thickness of the masonry. Stainless-steel corrugated sheets with side locks were used as insulation against rising damp.

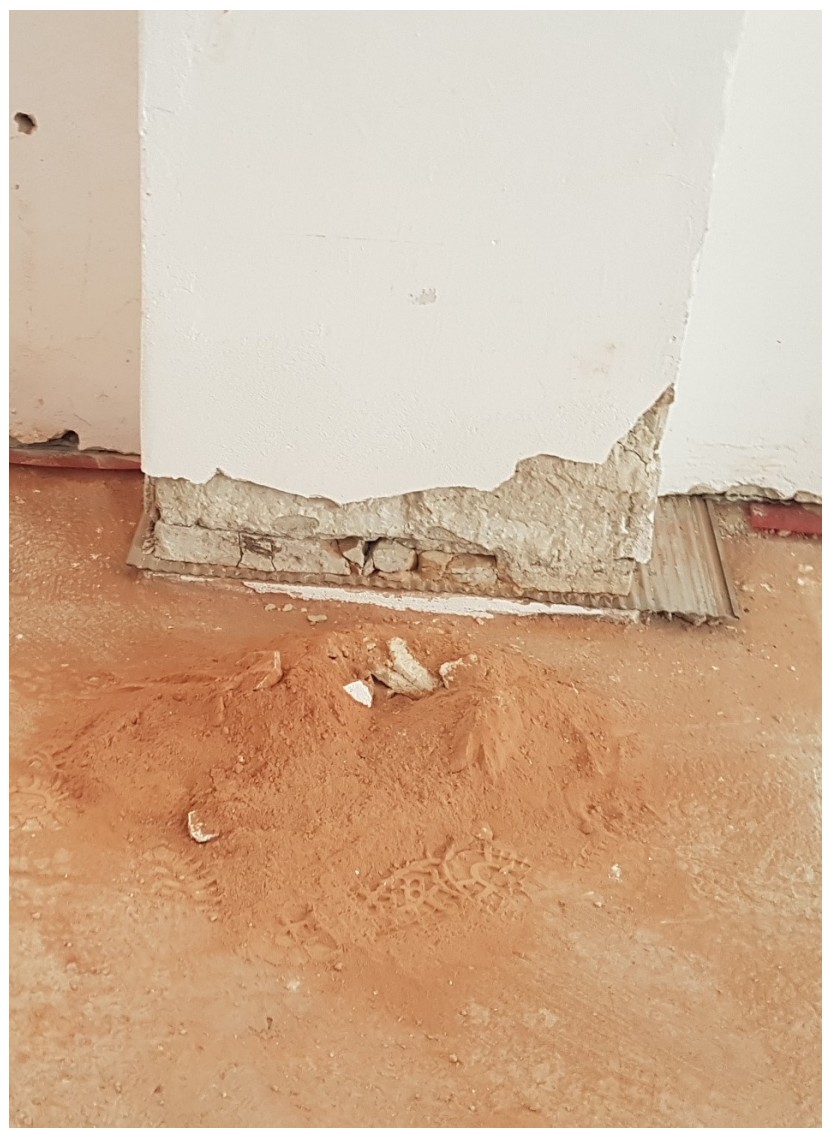

**Figure 4.** Application of corrugated stainless-steel plate stamping.

The third technology used was grouting (Figure 5). This was used in areas where access was not possible with the previous technologies. The other technologies could not be implemented due to the fact that it was not possible to undercut across the masonry at these locations, because the owner of the property renovating the indoor sanitary facilities in these locations. There were new tiles in these areas, as well as fittings, and in order to preserve them and to keep them intact, grouting was proposed. The individual implementation procedures are described in the following sections.

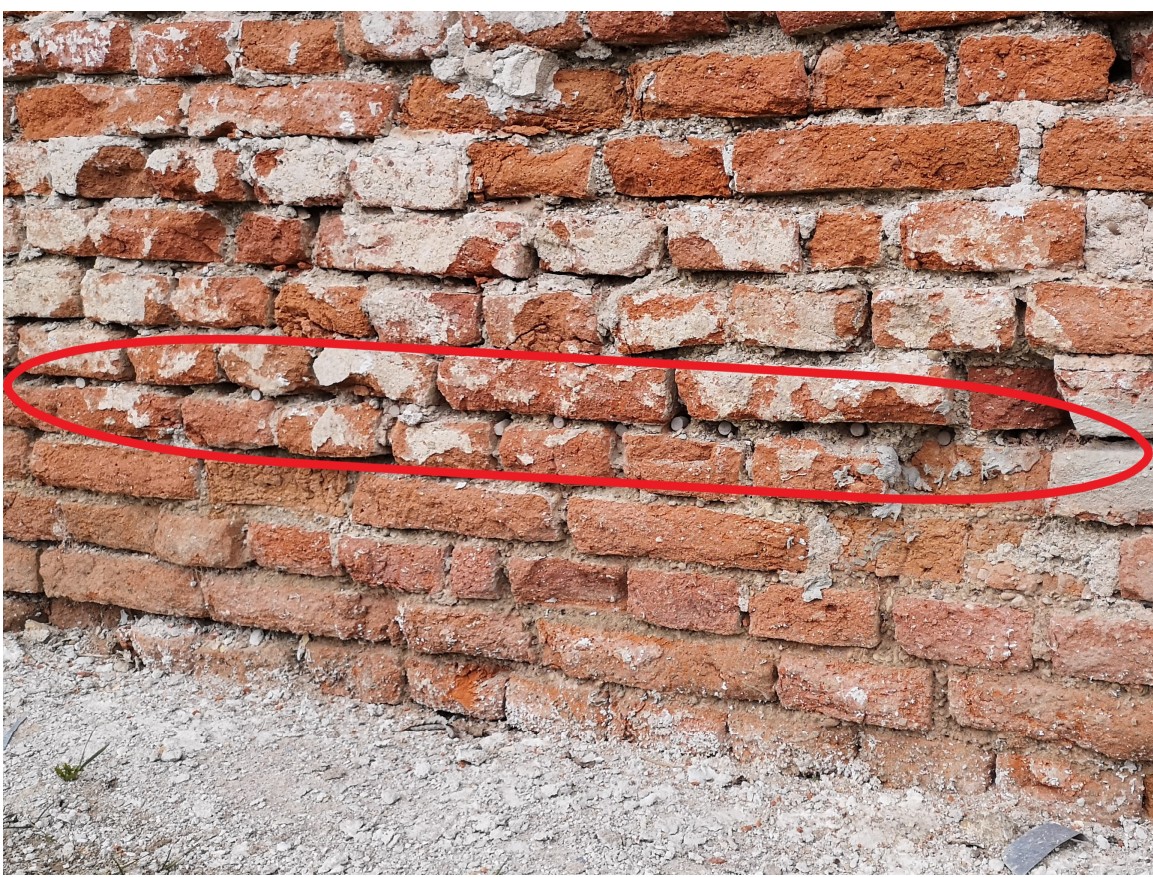

**Figure 5.** A grouting application line within the investigated building.

### 3.1. Implementation of Undercutting Technology

The primary technology deployed on the study site was undercutting technology. This was implemented in sequential steps using a diamond wire rope or, in exceptional cases, using a hand saw or chainsaw. The working span for the formation of the bedding joint was approximately 0.4 m long to ensure stability and safety. Subsequently, after undercutting (within the working width), the storage joint was cleaned. This procedure is important for the removal of mortar fragments and original masonry. If the joint had not been cleaned, it would not have been possible to ensure smooth insertion of the waterproofing strips into the joints. Cleaning was followed by the insertion of the waterproofing, and then the joints were lined with plastic anti-setting wedges. During implementation, attention was also given to the overlap of the individual waterproofing strips, which, in this case, was between 5 and 10 cm. In this procedure, the entire section of the building was undercut (except where another method was applied). Then, PVC pipes were fitted into the structure, and the structure was sealed at the surface with mortar. Once the mortar had hardened, the joint through the pipes was filled with expansion mortar. The individual sequential steps are presented in Figure 6.

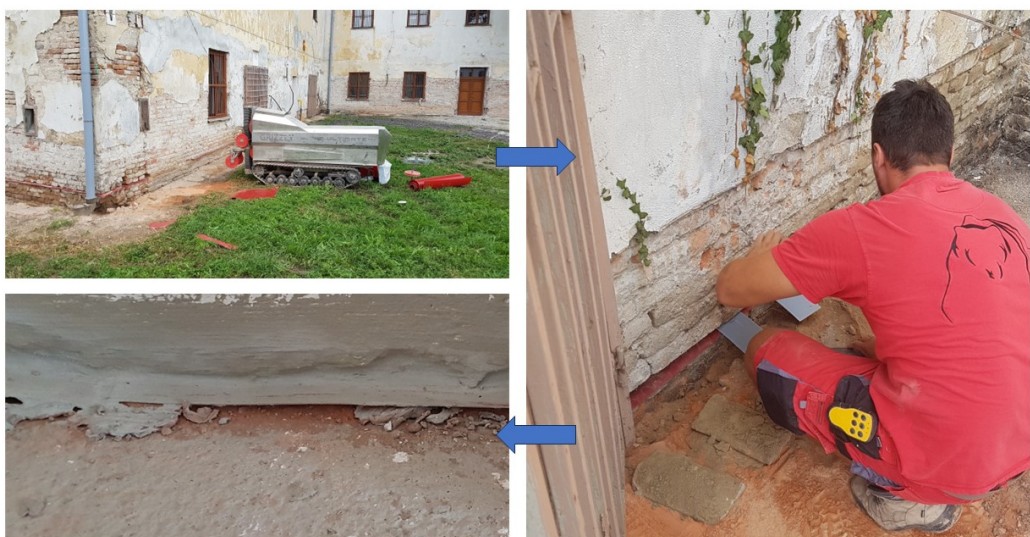

**Figure 6.** Sequential steps (abbreviated) of the implementation of the undercutting technology.

*3.2. Implementation of the Technology of Stamping Stainless-Steel Sheets*

In the area of the building where the masonry was crumbling, stamping technology was deployed in order to maintain the stability of the structure. This technology was used on a small scale, but for the purpose of completeness it is appropriate to describe its implementation. Additional insulation was implemented by embedding corrugated stainless-steel sheets into the masonry joints (Figure 7). These sheets were driven into the masonry by using a machine with compressed air. The insulation was grouted with a weaker displacement, but with a high frequency, i.e., rapid pressure vibration of the sheet at a frequency of 1200 beats/min.

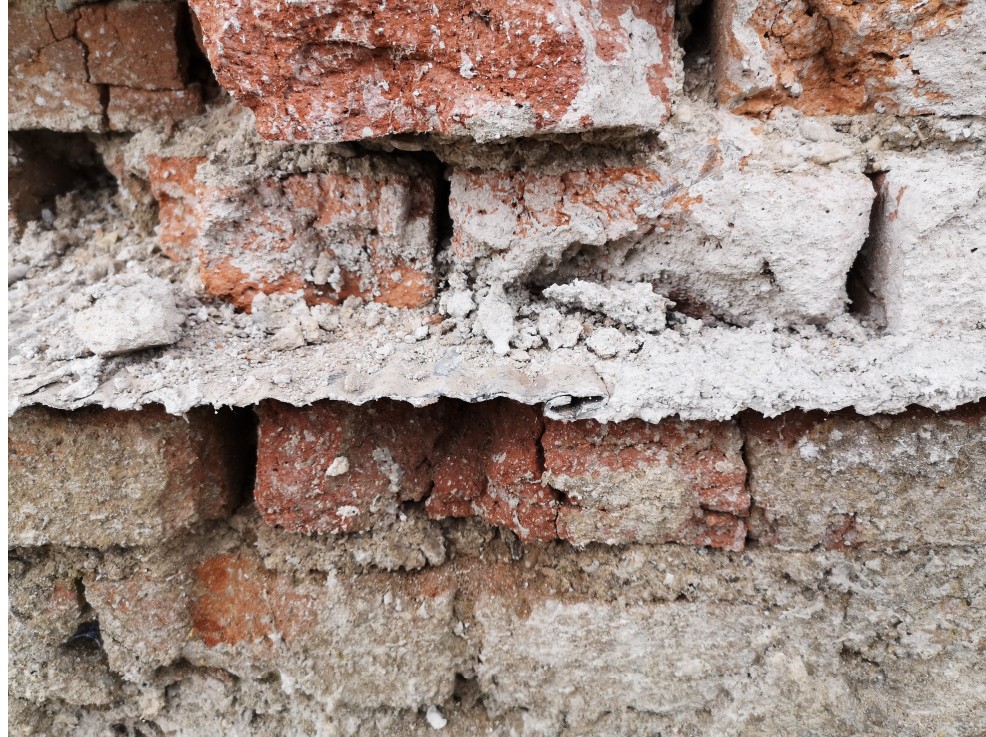

**Figure 7.** Slotted sheets applied to the buiding under observation. Detail of the lock, which ensures the waterproofness of the joints.

### 3.3. Implementation of Grouting

In the case of the building, grouting was carried out using a hydrophobic agent for remediation of wet masonry against capillary rising moisture using a system of drilled holes of the Remmers Kiesol C infusion screen (Figure 8). This substance can be characterized as solvent-free silane cream, and it is suitable for brick masonry forming a remediated structure.

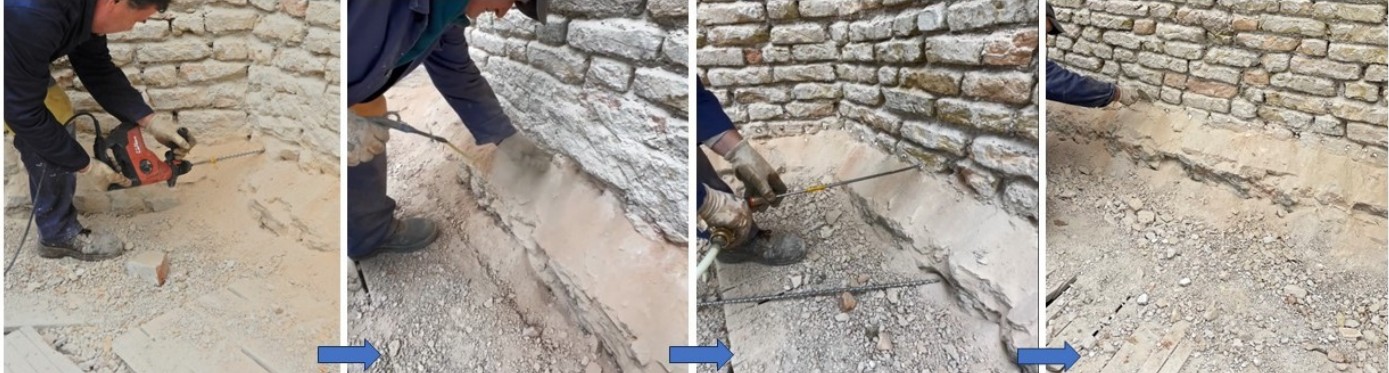

**Figure 8.** Grouting application procedure.

### 4. Research Methodology

The methodology of this research that verified the effectiveness of the above-described technologies applied on the historical Regional Monuments Office in Trnava, consisted of a long-term survey, which was carried out in situ. Therefore, it was possible to provide real results directly from the field that reflected the real situation, since various factors entered into the research that would not have been simulated in the laboratory and, from this perspective, the research results presented are authentic and can be applied directly in practice. Consequently, after evaluation, it is possible to determine the degree of effectiveness of the technologies investigated and their impacts in terms of sustainability ensured by the extension of the lifetime of the building.

The first step was a visual inspection of the building, with the aim of determining the current state of the building. At the same time, the level of dampness of the building was determined using a set of hygrometers that were deployed in random locations. These measurements provided a general idea of the degree of dampness in the walls of the building. For the in situ, dampness measurements, a Hygrometer BD2 contact moisture meter from Dosier Messgeräte, Füssen was used and, for the objectivity of the results, other devices were also used to determine the degree of dampness of the structure (hygrometer GANN, hygrometer TESTO 616). The characteristics of the individual devices and their parameters are given in Table 1. Using these instruments, a set of measurements was made with each instrument, which was then evaluated to eliminate as widely as possible the inaccuracies of the instrument measurements and the conditions during the measurements. Individual results were recorded in tables.

**Table 1.** Instruments used in the research (authors).

| Product Name | Power Supply | Measurement Range (%) | Depth of Measurement (mm) | Resolution (%) | Operating Temperature (°C) |
|---|---|---|---|---|---|
| Hygrometer Dosier DM4A-C | 9V | 0–20 | 40.00 | 0.10 | from 5 to 40 |
| Testo 616 | 9V | 0–20 | 50.00 | 0.10 | from 5 to 40 |
| GANN BL Compact B 2 | 9V | 0.3–11 | 120.00 | 0.10 | from 5 to 40 short term, from −10 to 60 |

Measuring the dampness content in the structure was the initial step to obtain information about the source of dampness, subsequent assessment and design of appropriate technology, and successful implementation of remediation intervention [26].

Several measurements were carried out to ensure the objectivity of the results. From local research destructive probes, it is clear that the building is constructed of brick masonry, namely solid burnt brick with a thickness of 52 cm. It is this material and its porosity that causes its high sorptivity, which implies that the dampness in the brickwork is due to capillarity.

The rate of damping of the building was investigated by performing measurements before and after the remediation interventions. The measurements were taken at predetermined points which were plotted on the floor plan of the building, and then they were repeated at specific locations. The measurements were carried out systematically at the marked points, using the three hygrometers listed in Table 1, and working on the principle of non-destructive measurement. Using these devices, a set of measurements was carried out using each device at each site. Then, the data were evaluated to eliminate, to the greatest extent possible, the inaccuracies of the instrument measurements and the conditions under which the measurements were made. Individual results were noted in tables. The exact measurement locations where dampness was measured throughout the survey had already been established during the first survey. This ensured the most accurate data possible, which could then be compared. As mentioned above, several sets of measurements were also taken after the implementation of the remediation measures, spaced over time. These measurements reflected the dampness pattern after their application. Finally, the results were compared, and the moisture level was assessed according to the Czech Technical Standard ČSN P 73 0610 [27], which classifies dampness into five groups (Table 2).

**Table 2.** Degree of moisture of constructions [13].

| | Degree of Humidity | Moisture ($\mu M$) (%) |
|---|---|---|
| 1 | Very low moisture | <3.0 |
| 2 | Low moisture | 3.0–5.0 |
| 3 | Increased moisture | 5.0–7.5 |
| 4 | High moisture | 7.5–10 |
| 5 | Very high moisture (to waterlogging) | >10 |

For better visualization and clarification, in Figure 9, the research methodology is elaborated as a flow chart, which simplifies the sequence of steps.

As described in the previous section, invasive remediation interventions were proposed for the subject property. The selection of the suitability of the deployment of invasive technologies was made on the basis of visual inspection, practical experience, and also after consultation and positive opinion of the Regional Monuments Office.

Despite the fact that these methods can be considered invasive and not very suitable from the point of view of the inappropriate interpretation of the Venice Charter [28] by some employees of the Monuments Office, these methods were applied to the building in this study. The design of the methods was also supported by previous practical experience with alternative technologies, such as venting technology [29], and their lack of effectiveness in the presence of high levels of dampness in the structure. Due to the high dampness, which can be considered to be waterlogging according to technical standards [27], a discussion was opened with the Monuments Office, which approved the subject proposal for the rehabilitation of the building. In the context of various studies that are not the subject of this publication, the authors often find themselves in discussions with the relevant monument conservation authorities about the appropriateness of deploying invasive technologies in the context of the restoration of historical buildings. Article 10 of the Venice Charter, which states the following, can be used to some extent as a justification for such interventions: Where traditional techniques prove inadequate, the consolidation of a monument can be achieved by the use of any modem technique for conservation and construction, the efficacy

of which has been shown by scientific data and proven by experience [28] According to this paragraph, the deployment of invasive remediation interventions could be seen as appropriate in the event of an unavoidable situation. It should be noted that, if implemented correctly, such techniques provide immediate long-term effects, at least in the area above the physical barrier [30], and thus prolong the service life of buildings. This effect is desirable from the point of view of sustainability and subsequent use of the building.

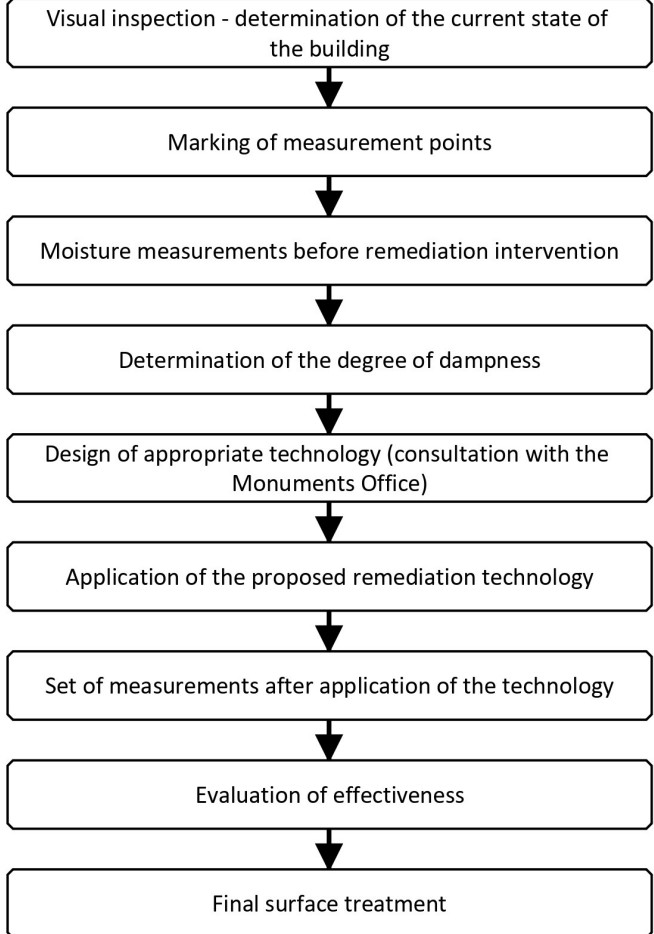

**Figure 9.** Flow chart describing the research methodology.

## 5. Description of the Building under Study

The building under study dates back to the 17th century and is located in the area of the National Cultural Monument Káčerov Majer. The buildings of the complex are located northwest of the historical core of the city of Trnava, directly at the flow of the Trnávka river, on its left bank [31]. Technically, the building is made of solid burnt brick. The brick walls of the building are approximately 52 cm thick. The plan shape of the building is L-shaped.

The area is named after one of the last owners, i.e., Henrich Káčer. The history of the whole area is very interesting, which includes the various uses of the building, whether for economic purposes or, later, as an army storehouse to a military school. Within these changes, there have also been structural interventions that have gradually changed the appearance of the building to its present state.

Originally, a hermitage with a garden stood on the site of the present site; however, it was often flooded by the aforementioned river. Subsequently, the property was sold, in 1776, to Michal Walter, who was a brewer. He built a representative residence and an outbuilding within the grounds. The oldest part of the depository building (the ground floor of today's southern part) was probably built in the first half of the 19th century and was used for economic purposes within the estate. Later, before 1895, it was rebuilt in

the shape of an L. Afterwards, the site had several owners, up to the above-mentioned Henrich Káčer. At the beginning of the 20th century, the site was used as a hussars' barracks, which means that, at that time, the premises already belonged to the state. Later, after the establishment of the First Czechoslovak Republic, the premises served as the School of the State Security Guard for Slovakia [31]. This resulted in the rebuilding, which involved the addition of the entire upper floor, during which the original roof trusses were sensitively preserved. This intervention gave the building its present form. Shortly thereafter it passed into the hands of the Czechoslovak army. Since 2002, the entire area of Káčerov Majer has become redundant property and is under the administration of the Monuments Office of the Slovak Republic.

The entire history of the building, especially its use for military purposes, has had a negative impact on the condition of the building. The building has been neglected, as can be seen in Figure 10.

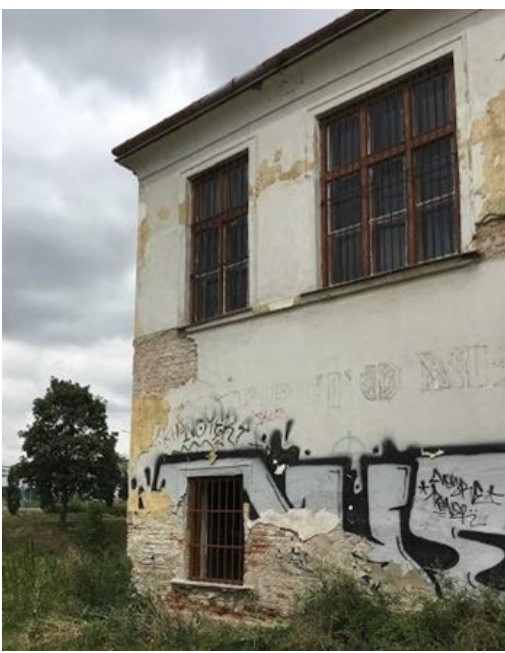

**Figure 10.** Detailed photograph of the degree of damage to the building caused by both dampness and salinity.

As can be observed from Figure 7, the building does not have a positive visual impact on the surrounding area. The dilapidated condition is also visible. During the initial inspection, several deficiencies of the building were found, such as a high level of dampness caused by rising damp. This had gradually attacked the various layers of the structure and, together with the water-soluble salts which had crystallized on the surface and formed efflorescence, was having a destructive effect on the plaster. The salinity levels were determined in aqueous leachate and were analyzed in the laboratory; however, the salinity level is not the subject of this study. From the results of the initial inspection and measurements, there was a need for a quick and radical solution to prevent the rising damp from having a destructive effect on the structure.

*Description of Site Weather Conditions*

To establish a general overview, it is also useful to examine the weather conditions of the site. Measurements were taken as part of the survey, but only on days when in situ surveys were carried out. For a general overview, the average weather conditions throughout the year are also given.

As part of the research, we used a meteogram which provided the necessary data to create an overview of the weather conditions of the area. A meteogram shows the annual

course of temperature, precipitation amount and days, as well as wind measurements from the closest weather station. The weather variables shown depended on the availability of complete and consistent measurements from the measurement station and were aggregated on a weekly or monthly basis [32]. The above described is shown in Figure 11.

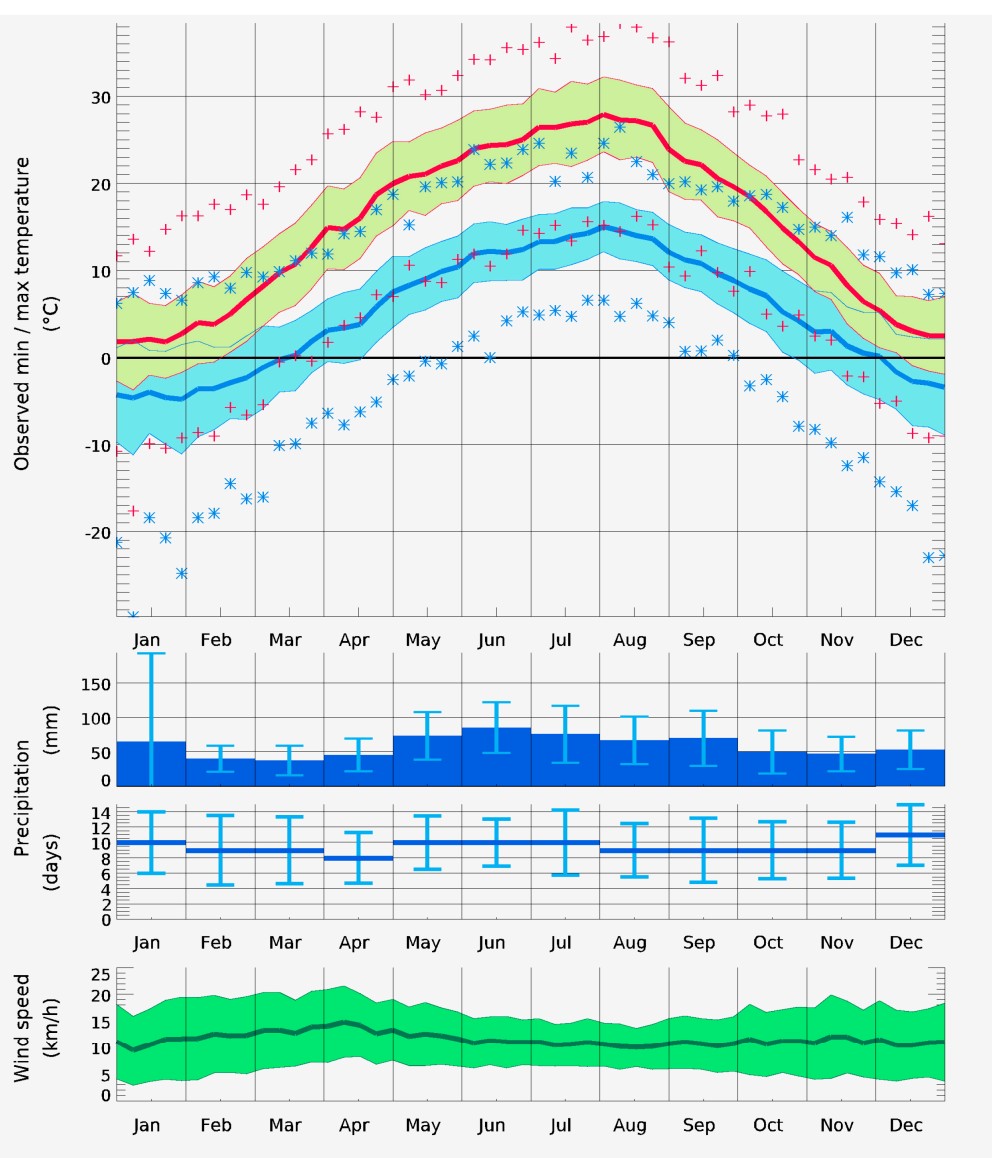

**Figure 11.** History of weather conditions in the vicinity of the building under study [32].

The first diagram shows the average, the maximum (red), and the minimum (blue) temperatures. Extreme values are represented by + and * characters. Two-thirds of the observed temperatures lie within the colored temperature range. The second diagram shows the precipitation amount (in mm) and the range of monthly means in two-thirds of the years. The third diagram shows the number of days per month with precipitation and the two-thirds variation within the bars. The last diagram (No. 4) shows the observed daily mean wind speed and the two-thirds range of weekly means [32].

## 6. Research Results

As described in more detail in the previous section, various changes, both in layout and especially in purpose, have had an impact on the condition of the building. The building was in a state of disrepair. The exterior and interior plasterwork was not functional and,

visually, the building did not look satisfactory. The condition was mainly due to the high dampness caused by rising damp. A request was made to restore this building for new use as a depository. For this reason, research was carried out to measure the dampness, and then to design a suitable technology to remove it. The different measurement locations and technologies used within the building under investigation are represented in Figure 12. The rate of dampness is shown in Table 3. The research was carried out between 2017 and 2020.

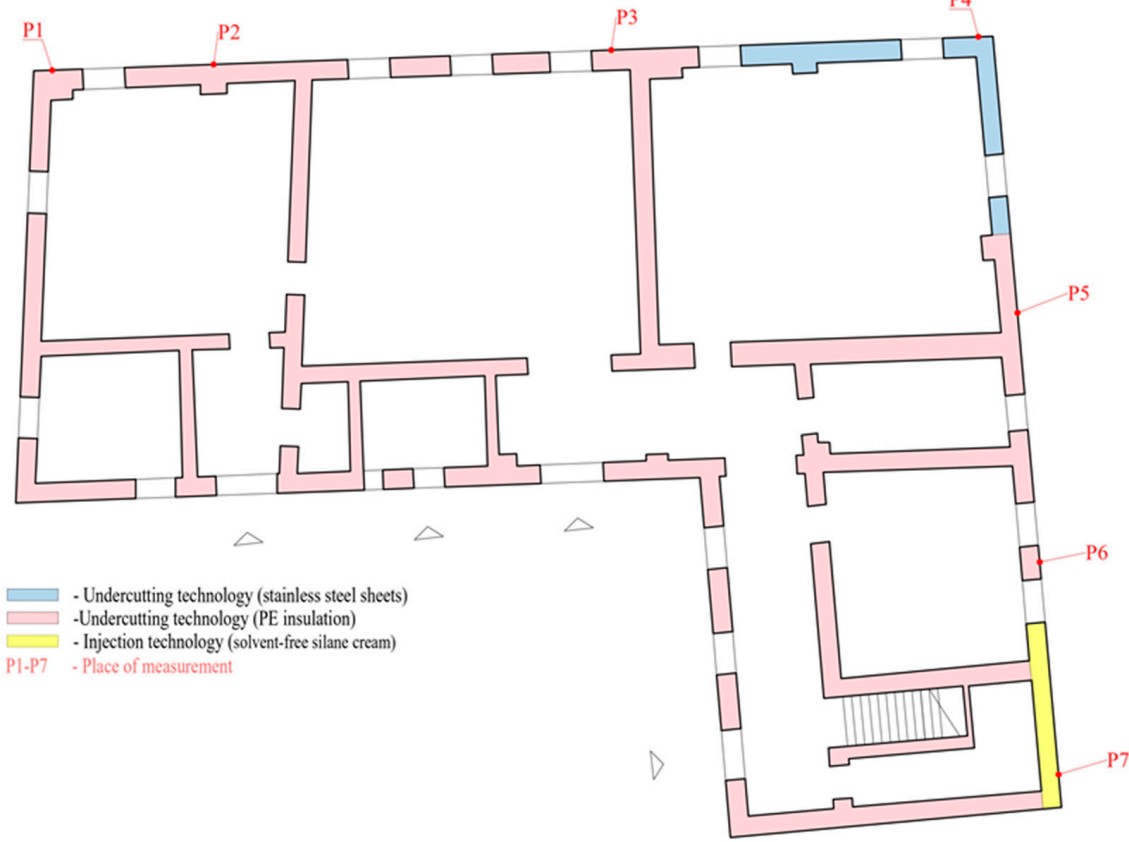

**Figure 12.** Moisture measurement points and indication of the range of applied technologies.

**Table 3.** Results of dampness before and after the application of remedial interventions.

| Place of Measurement | Measuring Height from Floor (cm) | Mass Moisture (%) | | | Measuring Height from Floor [cm] | Mass Moisture (%) | | | Comment |
|---|---|---|---|---|---|---|---|---|---|
| | | 7.7. '17 | 8.11. '19 | 11.3. '20 | | 7.7. '18 | 8.11. '19 | 11.3. '20 | |
| P1 | 30 | 17.3 | 1.4 | 2.3 | 150 | 2.2 | 2.1 | 3.4 | F [1] |
| P2 | 30 | 12.5 | 1.7 | 1.9 | 150 | 8.7 | 6.5 | 6 | F [1] |
| P3 | 30 | 9.2 | 3.4 | 2.1 | 150 | 2.7 | 1.8 | 1.4 | F [1] |
| P4 | 30 | 16.6 | 1.6 | 1 | 150 | 11.8 | 7.7 | 5.2 | F [1] |
| P5 | 30 | 14.3 | 2 | 1.8 | 150 | 6 | 2.2 | 2.2 | F [1] |
| P6 | 30 | 10.9 | 2 | 2.3 | 150 | 4.1 | 2.4 | 2.5 | F [1] |
| P7 | 30 | 5.3 | 1.9 | 1.4 | 150 | 6.3 | 6 | 7.2 | F [1] |
| $TAir_{ext}$ [2] (°C) | | 27.5 | 14 | 13.1 | $TAir_{int}$ [3] [°C] | 26.4 | 12.6 | 12.3 | |
| $\Phi_{ext}$ [4] (%) | | 69 | 70 | 66.8 | | | | | |
| $TWall$ [5] (°C) | | 22.4 | 19.5 | 18.9 | | | | | |

[1], façade; [2] Exterior air temperature; [3], interior air temperature; [4], humidity of the air in the exterior; [5], wall temperature.

The initial measurements were performed throughout the entire building. Due to the COVID-19 pandemic and also due to the progress of work by the contractor, who

proceeded to complete the remediation plaster and surface finishes after the implementation of the remediation interventions, it was not possible to measure the effectiveness of these technologies at other points. We were invited only after the application of the surface finishings in the other areas, which made the other measurements difficult; however, these points covering all the applied technologies are sufficient to form a complete image. The survey was always measured at the indicated locations using three different measuring devices, the specifications of which are given in Table 1. In this way, it was possible to eliminate measurement inaccuracies and determine the most accurate dampness values at each specific point of the structure under investigation.

The dampness in the surveyed property after the initial survey was determined to be severe (waterlogged in many areas). The building exhibited high levels of waterlogging, which contributed significantly to its destruction and to the impossibility of further use. Consequently, remediation proposals were drawn up and implemented. Approximately 11 weeks after the remediation interventions, when the wall was deprived of direct contact with the wet subsoil by these measures, the structure was allowed to dry out. Subsequently, with an interval of 4 months, the last measurements were conducted before the application of the surface layer (remedial plaster).

We recorded the individual values, and then listed the resulting values in a table that is included in this paper and shown above. The results were also depicted in graphs (see Schemes 1 and 2) that represent the dampness trends over time before and after the implementation of the damp-proofing interventions. There are two graphs, one graph shows the dampness content measured at 30 cm above ground level and the second graph represents the dampness and its progression at 150 cm above ground level.

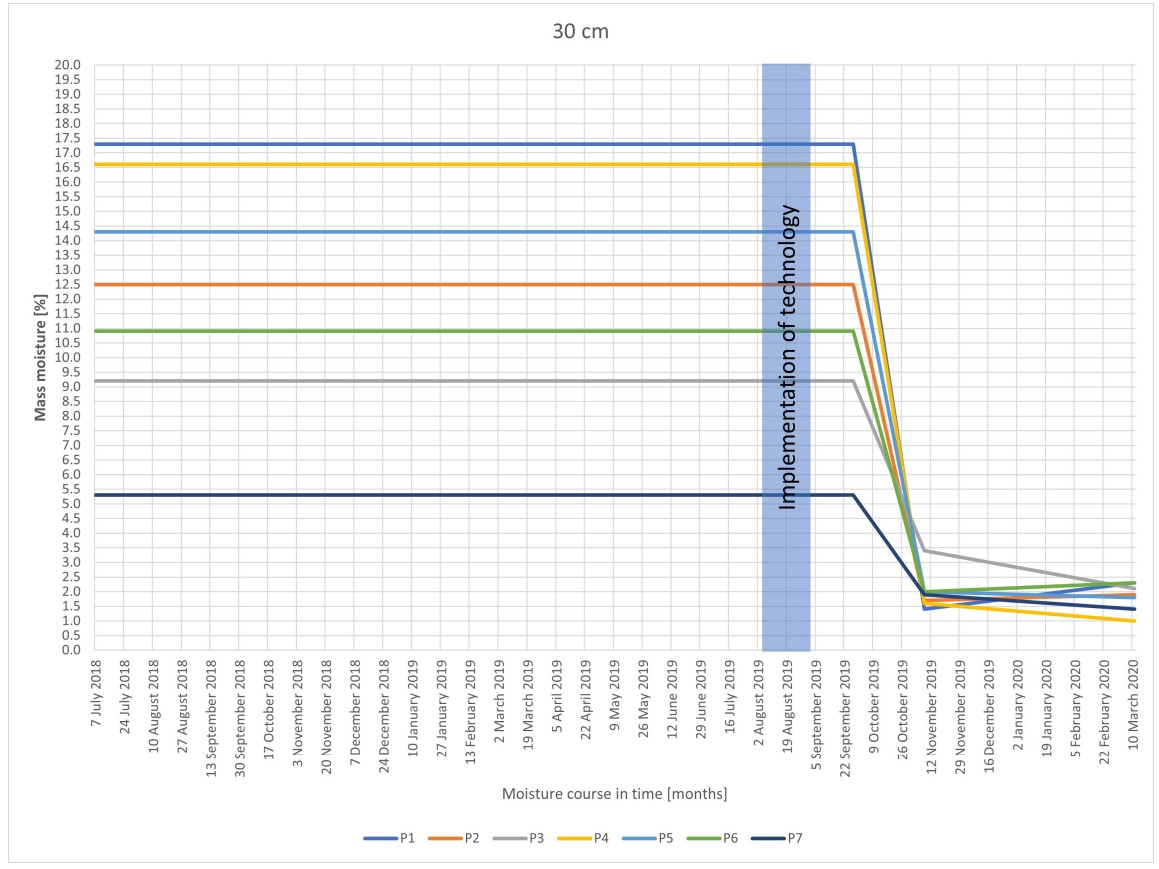

**Scheme 1.** Graph showing the dampness progression of the structure before, during, and after implementation of the damp-proofing technologies measured at a height of 30 cm.

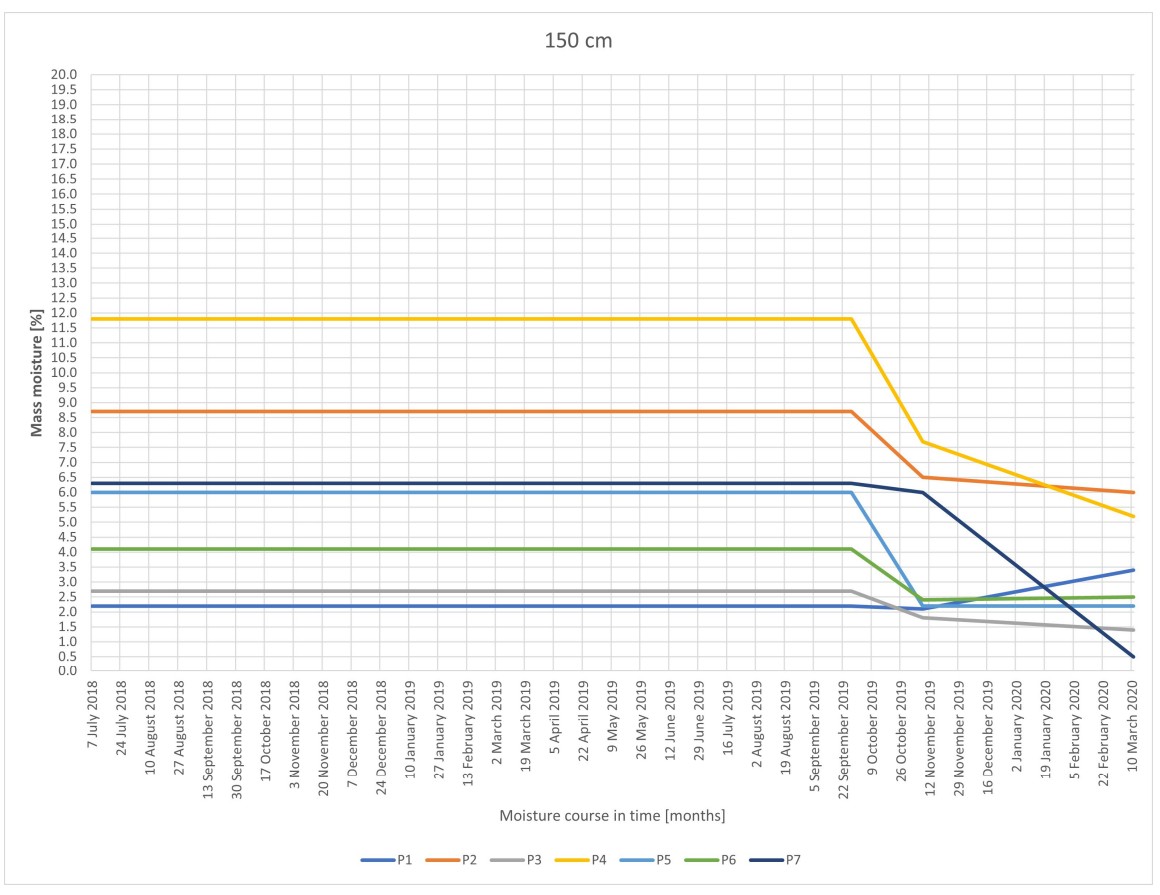

**Scheme 2.** Graph showing the dampness progression of the structure before, during, and after implementation of the damp-proofing technologies measured at a height of 150 cm.

Based on the measurements, it is certain that, in the areas just above the ground where the highest humidity was concentrated according to the first measurements, it had receded considerably. When assessed according to the technical standards [13], it can be concluded that the structure had dried out, as demonstrated by the results in Table 4, which describe the individual dampness structure before and after remediation, dividing the results according to the technology and the measurement height. The table also includes a percentage assessment of the dampness loss.

**Table 4.** Display of the state and percentage loss of moisture in the investigated building (authors).

| | Káčerov Majer | |
|---|---|---|
| | **30 cm** | **150 cm** |
| Average humidity before remediation (%) | 12.30 | 5.97 |
| Average humidity after remediation (%) | 1.83 | 3.03 |
| % difference before and after rehabilitation (%) | −85.13 | −49.28 |
| Average humidity before remediation EXT [1] (%) | 9.14 | |
| Average humidity after remediation EXT [1] (%) | 2.43 | |
| % difference before and after rehabilitation (EXT) [1] (%) | −73.42 | |
| Masonry type | Brick | |
| Undercutting PE ins. [2] (% moisture reduction) (%) | −54.20 | |
| Undercutting stainless steel sheets (% moisture reduction) (%) | −74.95 | |
| Injection technology (% moisture reduction) (%) | −82.82 | |

[1] EXT, exterior; [2] PE ins., polyethylene insulation.

The positive effects of the interventions can also be seen in Figure 13, which shows before and after photographs of the building under study. It is also worth noting that the building has found a new use and serves as a depository for the Regional Monuments Office in Trnava.

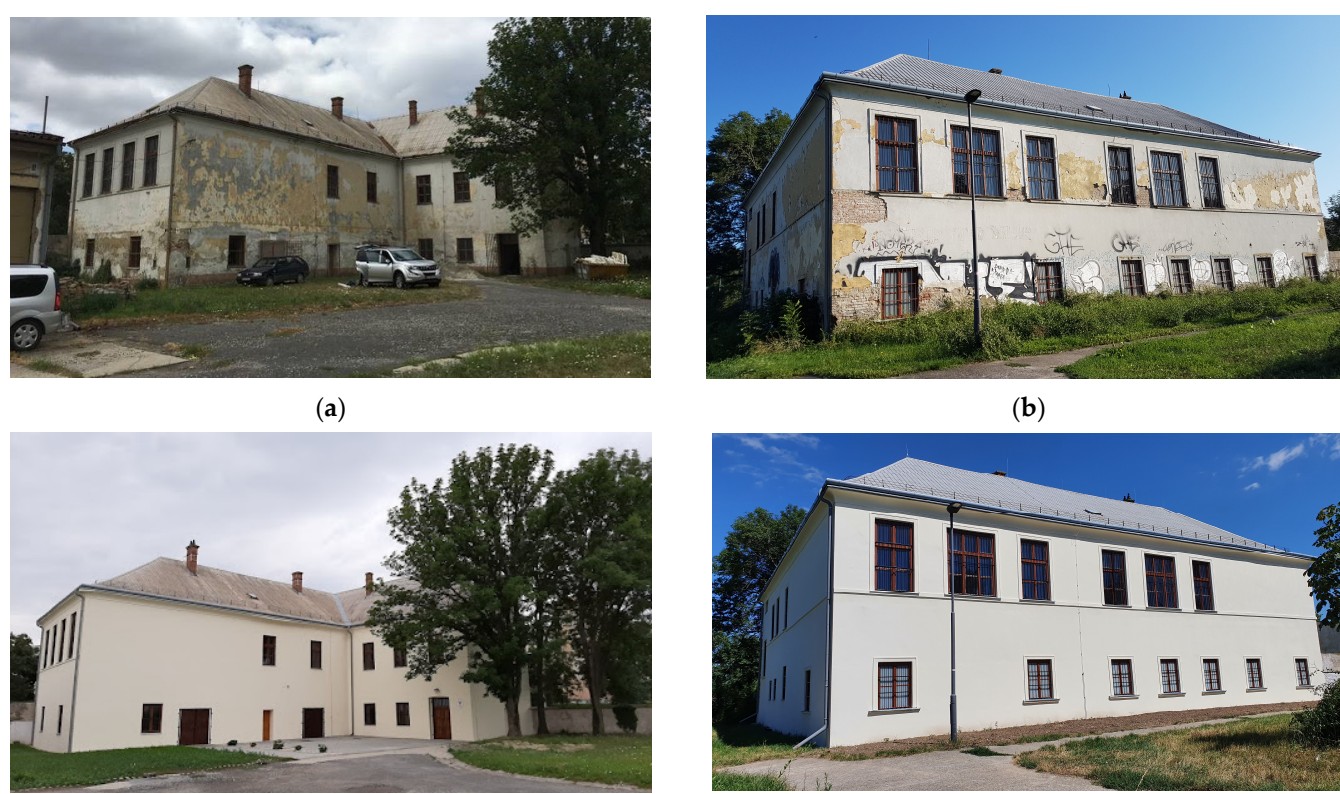

**Figure 13.** View of the state of building: (**a**,**b**) Before; (**c**,**d**) after recovery.

## 7. Discussion

As described in the previous sections, the technologies applied to the building in question are in many cases controversial from the point of view of monument preservation authorities; however, there are cases where they are necessary, especially in the case of high levels of dampness caused by rising damp. It is likely, based on additional research, that many non-invasive technologies may not cope with high dampness [33,34].

On the contrary, as also mentioned in the overview and state of the art, the technologies that were the subject of this research have been applied on several sites and their effectiveness has been verified [22,35,36]

The different techniques of remediation interventions in this research were chosen mainly because of the complexity of the structure and the surrounding conditions. Grouting was applied in areas where it was not possible to apply stainless-steel plate stamping and undercutting technology. Stamping technology was chosen in areas where the original masonry was crumbling and undercutting could lead to its complete destruction. In these areas, sheet metal also performed a reinforcing function and the crumbled part above the level of the insulation was replaced. However, the disadvantage of this technology is the possible problematic contact between the sheets, through which dampness can continue to rise if executed improperly. In other areas, undercutting technology was used and applied to the largest part of the building. The advantage of applying one technology to the largest possible extent is uniformity of execution, which subsequently affects price and quality. The disadvantage is the large working space required for the execution. The scale and scope of technology deployment is represented in Figure 9.

Based on research and analysis, it can be concluded that the interim results demonstrate the effectiveness of the technologies used to prevent the penetration of capillary dampness through the application of additional impermeable layers, as evidenced by the measured values described in the individual tables. This demonstrates that the use of such technologies reflects positively on the condition of the building under study in terms of its sustainability.

Evidently, the results can also be compared with other research dealing with these technologies. Together with these findings, we can draw conclusions that are relevant and can be followed up with further research.

Although the remediation interventions consisting of the above-mentioned technologies may seem radical and costly, it is indeed positive in the long term in relation to the environmental, financial, and technical sustainability of the building. The application has provided a functional and healthy environment that has contributed to the reuse of the building, which does not burden the environment with its degradation and does not require additional financial costs associated with the partial maintenance of the original condition.

The results of this study should assist in expanding the knowledge of appropriate remediation technologies and their benefits in terms of long-term sustainability.

It should be noted that it is not necessary to implement the reviewed methods on all constructions and it is always necessary to approach each construction individually. There are existing and ongoing studies dealing with the level of effectiveness of non-invasive interventions and consideration should also to be given to these technologies and their level of effectiveness.

## 8. Conclusions

From the results, it is apparent that the subject technologies applied to the building under study show functionality and the necessary efficiency. Based on the data, the technologies can be assessed as very positive in terms of extending the lifetime of the building and also its further long-term use. This can be considered to be a very significant contribution to the sustainability of the building. For example, from a financial point of view, the building will not require the renewal of any parts of the building that were previously dilapidated and in a state of disrepair for several years, which can be considered to be economic sustainability. This follows sustainability from an environmental point of view, where dilapidated parts will not create debris and waste around the building, which has a positive impact on the environment. The third aspect related to the sustainability of the building under study is the new use of this building as a depository, due to the new dry environment created once the waterlogged walls inside have dried out.

Applied technologies are created at the cost of disturbing the original integrity of an historical structure. This is often not appropriate from the point of view of restoration and conservation. In such a case, two different views of heritage buildings stand in opposition to each other. Either the buildings will dry out even at the cost of invasive interventions or dampness will continue to have a destructive effect on the buildings. The dampness of structures will continue to be coupled with the humidity of the environment, which will have a negative effect on the people using the buildings, or it will have a negative and destructive effect on the various equipment found in this environment (furniture, books, artifacts, paintings, etc.). The justification for a positive attitude towards the implementation of such interventions is also supported by the results of several experiments showing the functionality of invasive methods as opposed to many non-invasive methods that often only provide functionality to a certain extent and cannot handle high levels of waterlogging. An example is the magnetokinesis technology, which has become popular in recent years. This is used extensively in Slovakia in churches and various historical buildings. The rationale for its deployment is mainly that it does not require any interferences in the structures. However, the opposite is true. Authors are currently working on this topic and there are already several publications. The contractors refer to this method as functional and non-invasive, but its functionality is mainly ensured by the design of the new remedial

plaster, which is applied to a height of about 1.5 m from the floor and subsequently indicates the dryness of the surface when measurements are performed. Similar methods are also financially demanding, and their non-functionality has been demonstrated in most cases after a long period of time when the applied remedial plaster has lost its functionality. Therefore, such methods cannot be applied to buildings where we want to achieve their reuse and prolong their lifetime in terms of long-term sustainability.

It is probable that if the possibility of implementing the technology of additional impermeable layers on selected heritage-protected buildings, which are heavily exposed to rising damp, is not taken up, their technical damage and, in worse cases, their complete destruction may occur. This is unacceptable from the point of view of the protection of monuments as a form of cultural heritage. At the same time, such buildings must also be viewed from the point of view of future generations, for whom they must be protected and maintained.

We understand that the use of radical interventions cannot be applied to all buildings affected by dampness. However, it is clear that such interventions are effective and, in selected cases, necessary to ensure the sustainability of the building from various aspects. It can also be a challenge for the future elaboration of a methodology that would find an optimal solution suitable for conservation authorities and the owners of such buildings. In situ studies, laboratory analyses, as well as the experience of experts in the field should be taken into consideration in their development. Such a methodology would clearly define when and under what conditions individual technologies may be implemented, to what extent, and on which sites. Such rules and procedures would make a positive contribution to the long-term sustainability of these buildings, which could be restored to their function or find other uses. Consequently, for selected building types, it is possible to focus on their indoor climate and environment and to propose appropriate measures related to the design of heating and cooling sources to optimize the use of the building.

**Author Contributions:** Conceptualization, P.Š. and I.V.; methodology, P.Š, P.M., I.V., and D.K.; software, P.Š.; validation, P.Š., P.M., I.V. and D.K.; formal analysis, P.Š., P.M., I.V. and D.K.; investigation, P.Š.; resources, P.Š. and I.V.; data curation, P.Š. and I.V.; writing—original draft preparation, P.Š., P.M. and I.V.; writing—review and editing, P.Š. and I.V.; visualization, P.Š.; supervision, P.Š., P.M. and D.K.; project administration, P.Š.; funding acquisition, P.M. All authors have read and agreed to the published version of the manuscript.

**Funding:** This research is supported by the Ministry of Education, Science, Research and Sport of the Slovak Republic through the grant VEGA 1/0118/23.

**Institutional Review Board Statement:** Not applicable.

**Informed Consent Statement:** Not applicable.

**Conflicts of Interest:** The authors declare no conflict of interest.

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
