# Peer review of "Implementing the Technologies of Additional Impermeable Layers in a Building of the Monuments Office (Káčerov Majer) from a Sustainability Point of View"

_sustainability, doi:10.3390/su151411187_

Round 1

Reviewer 1 Report

The herein study, i.e., " Implementation of the Technology of Additional Impermeable Layers on the Building of The Monument Office (Káčerov Majer) from a Sustainability Point of view", submitted to the Journal Sustainability, is relevant to the body of knowledge on materials to solve problems of rising dump in historical buildings.

The English of the work needs to be reviewed.

The use of the technologies is stated as a sustainable solution but is not shown why.

Has this type of technology never been used? Reviewing these techniques on previous works is essential for the work (part of this revision is done in chapter five, but should be in the introduction or a chapter of state of the art).

The methodology should show all the steps that need to be done to obtain a final solution.

What measures were done, with which periodicity, and in what conditions? Go deeper into this description of the work done.

Several solutions were applied. How was the choice made?

Some pictures taken during the procedures would be significant.

Line 63 – What is WTA? Also, refer to the standards.

Before chapter 4, the building should be described. The need for rehabilitation and the description of the problems should be stated. Also, the weather conditions of the place should be described (Ambient temperatures during the year, solar hours, rainfall, wind direction and velocity…)

Table 3 – units

The discussion is vague and would benefit from a state-of-the-art review of the techniques usually used for similar situations—the advantages and disadvantages of each application. The reasons why you choose the techniques for each place of the building would be necessary for the work.

Line 214 – Which substances?

Line 217 – The results or a summary of the laboratory results should be presented even though they differ from the ones In Situ. If it is not important at all, remove them from the paper.

Line 219 – How was it directly confirmed?

Reviewer 2 Report

The prevention and control of rising damp in cultural heritage is a research topic with a long history. Since the middle of the 20th century, different scientists have tried to solve the problem of rising damp by different methods such as physics, chemistry, mechanics and even electricity, and although they have achieved good results, they have also come to the common view that the actual situation must be respected.

Main comments are as followed,

1. The author seems to have misunderstanding about the preservation of the Monument, citing the Charter of Venice, but not clear about the intrusive restoration method and the conflict between the Charter.

2. There is no direct relationship between the article and the idea of sustainability point of view in the title.

3. Misunderstanding moisture and rising damp.

4. There are big doubts about the structure of the article. The discussion part is similar to literature review. It is appropriate to discuss this part at the front of the article.

5. The content of the conclusion is not highly relevant to the article.

6. To sum up, the concept and structure of the full text should be rearranged.

Reviewer 3 Report

Comments and Suggestions for Authors

Researchers should take the following research notes into consideration:

·         The research is satisfactory in terms of scientific originality and the breadth of its scientific significance in the field, and the study's objectives were met. The title accurately reflects the content of the paper, and the research concept is distinct. The researchers employed a semi-clear and validated methodology in the research structure. The paper makes a significant literary contribution. The conclusions are comprehensible, correctly interpreted, and supported by the data, which were presented in a clear manner. The writing is clear, concise, and well-organized in the manuscript. The abstract is succinct and lists the key ideas of the paper. The paper's relevance to the field is explained in the introduction, which also offers a critical review of recent literature and identifies the areas in which the paper is relevant.

·         The manuscript is coherent, well-written, and well-structured. The abstract could use some improvement but it is succinct and summarizes the key ideas of the paper. The introduction gives a critical analysis of the available literature, identifies the areas in which the paper is pertinent to the field, and explains how the paper is pertinent to the field but that there are some factors that need to be taken into account.

1.       abstract: The methodology and key research findings should be more accurately described in the abstract.

2.       Key words: Some Keywords related to title and location need to added, such as Kácerov Majer area sustainability.

3.       Introduction: The important terms should be included in the introduction:

·         A section devoted to the documents and methodologies of heritage restoration processes, such as the concept of reinstatement and the Venice Charter, their significance and how to work with them, as referred to by the researchers in the research methodology and results, and thus this section and its significance must be clarified in the theoretical introduction.

·         A section devoted to explaining and defining the Kerov Majer case study, as well as its history and reality

·         I recommend that the proposed paragraphs be added to the theoretical section to strengthen it. Attached are some updated reference suggestions for the theoretical introduction, as well as Materies and Methods to expand the theoretical framework of research.

o    ENVIRONMENTAL MANAGEMENT A DOORWAY TO GREEN INVESTMENT OF HERITAGE ENVIRONMENTS”

o    Green Investment of heritage Environments: A doorway to improving quality of life http://dx.doi.org/10.2139/ssrn.3163512

o    Restoration Towards sustainable green heritage buildings, Case Study: Mansoura Opera House, Mansoura, Egypt, http://dx.doi.org/10.21608/SJFA.2018.203494  

4.       Materials and Methods: To make it shorter and clearer, I advise putting the methodology clarification in flowchart of the steps that were taken at each stage.

5.       Results: Why was only the other side of the building (north and east) satisfied with the humidity measurement, as shown in Figure 4 from the entrance side (west and south)? Is there a rationale behind that?

6.       Discussion: I'm mentioned by the authors on line 236.” the author lacks the continuous measurement of humidity after the application of the technology to establish certain conclusions.” What is the outcome, and how does that affect the findings of the research?

7.       Conclusions:  This section requires more clarification to understand the Venice Charter and the procedures that can be done and clarified during building restoration, as it is possible to use a number of procedures with reference to and clarification historically in a way that does not affect the building's safety, and thus this must be clarified, and the extent of understanding of this should be indicated in the research recommendations.

8.       References: In general, the number of references is adequate, and most of them are recent and well written. The relevant and necessary references have been included, were cited correctly throughout the paper.  

Round 2

Reviewer 1 Report

The authors made most of the proposed changes, the writing needs a little revision in the final format.

Reviewer 3 Report

I appreciate you checking to see that many of the necessary revisions were completed, and I wish you luck.